# Identification and Characterization of ATOH7-Regulated Target Genes and Pathways in Human Neuroretinal Development

**DOI:** 10.3390/cells13131142

**Published:** 2024-07-03

**Authors:** David Atac, Kevin Maggi, Silke Feil, Jordi Maggi, Elisa Cuevas, Jane C. Sowden, Samuel Koller, Wolfgang Berger

**Affiliations:** 1Institute of Medical Molecular Genetics, University of Zurich, 8952 Schlieren, Switzerland; grubichatac@medmolgen.uzh.ch (D.A.); kmaggi@medmolgen.uzh.ch (K.M.); feil@medmolgen.uzh.ch (S.F.); maggi@medmolgen.uzh.ch (J.M.); koller@medmolgen.uzh.ch (S.K.); 2UCL Great Ormond Street Institute of Child Health, University College London and NIHR Great Ormond Street Hospital Biomedical Research Centre, London WC1N 1EH, UKj.sowden@ucl.ac.uk (J.C.S.); 3Zurich Center for Integrative Human Physiology, University of Zurich, 8057 Zurich, Switzerland; 4Neuroscience Center Zurich, University and ETH Zurich, 8057 Zurich, Switzerland

**Keywords:** ATOH7, retinal organoids, retinal development, retinal ganglion cells, retinal progenitor cells, RNA sequencing, scRNA sequencing, CUT&RUN sequencing

## Abstract

The proneural transcription factor atonal basic helix–loop–helix transcription factor 7 (*ATOH7*) is expressed in early progenitors in the developing neuroretina. In vertebrates, this is crucial for the development of retinal ganglion cells (RGCs), as mutant animals show an almost complete absence of RGCs, underdeveloped optic nerves, and aberrations in retinal vessel development. Human mutations are rare and result in autosomal recessive optic nerve hypoplasia (ONH) or severe vascular changes, diagnosed as autosomal recessive persistent hyperplasia of the primary vitreous (PHPVAR). To better understand the role of *ATOH7* in neuroretinal development, we created *ATOH7* knockout and eGFP-expressing *ATOH7* reporter human induced pluripotent stem cells (hiPSCs), which were differentiated into early-stage retinal organoids. Target loci regulated by ATOH7 were identified by Cleavage Under Targets and Release Using Nuclease with sequencing (CUT&RUN-seq) and differential expression by RNA sequencing (RNA-seq) of wildtype and mutant organoid-derived reporter cells. Additionally, single-cell RNA sequencing (scRNA-seq) was performed on whole organoids to identify cell type-specific genes. Mutant organoids displayed substantial deficiency in axon sprouting, reduction in RGCs, and an increase in other cell types. We identified 469 differentially expressed target genes, with an overrepresentation of genes belonging to axon development/guidance and Notch signaling. Taken together, we consolidate the function of human *ATOH7* in guiding progenitor competence by inducing RGC-specific genes while inhibiting other cell fates. Furthermore, we highlight candidate genes responsible for *ATOH7*-associated optic nerve and retinovascular anomalies, which sheds light to potential future therapy targets for related disorders.

## 1. Introduction

The seven main cell types in the human retina, including retinal ganglion cells (RGCs), horizontal cells, amacrine cells, bipolar cells, cone and rod photoreceptors (PRs), as well as Müller glia, all arise from multipotent retinal progenitor cells (RPCs) in an overlapping but conserved temporal order [1]. The shifting landscape of competence in RPCs is orchestrated by both cell extrinsic and intrinsic factors, the latter including a set of neurogenic and proneurogenic developmental factors belonging to the family basic helix–loop–helix (bHLH) transcription factors, which are involved in regulating proliferation and differentiation through, among other mechanisms, association with Notch signaling [2,3].

A significant body of evidence suggests that RGCs, the first-born retinal neurons, require the expression of the atonal basic bHLH transcription factor 7 gene (*ATOH7*). Loss-of-function mutations in various in vivo models present with an almost complete absence of RGCs [4,5,6,7,8,9]. In addition to model animals, patients with mutations in *ATOH7* or its remote regulatory enhancer element display optic nerve hypoplasia (ONH), as well as aberrations in the retinal vasculature, including persisting hyaloid vasculature, incomplete retinal vascularization, retinal neovascularization, and the ensuing tractional detachment of the retina [10,11,12,13,14,15,16,17]. These changes are believed to be secondary to RGC loss and may be a result of loss of paracrine factors and/or cell-to-cell interactions. In severe cases when retinal detachment is complete before birth, these changes are described as autosomal recessive persistent hyperplasia of the primary vitreous (PHPV; OMIM: #221900), sometimes also referred to as nonsyndromic congenital retinal non-attachment (NCRNA), or as persistent fetal vasculature (PFV) [18,19]. A highly related condition that *ATOH7* has also been associated with is familial exudative vitreoretinopathy (FEVR; OMIM: #133780, #601813, #616468), where the vascular changes are usually milder and the retinal detachment, instead, presents as incomplete and progressive [20,21]. Thus, FEVR and PHPV can be seen as a spectrum, with most severe cases presenting as PHPV [14].

The exact molecular mechanisms mediated by ATOH7 in retinal development and associated pathologies have for a long time been elusive. In particular, the activation of *ATOH7* expression in the multipotent progenitor does not irreversibly specify its fate but rather favors RGC competence. Fate mapping indicates that *Atoh7*-expressing RPCs commit to all retinal fates, with a strong preference towards the earlier-born cone PRs and amacrine and horizontal cells, substantiated by a fate shift towards these cell types in *Atoh7*-deficient retinae [22,23,24,25]. Additionally, birth dating and lineage tracing suggest that approximately half of all new-born RGCs are dependent on *ATOH7* expression [22]. Recently, a study was conducted on *Atoh7*-knockout (KO) mice, which were simultaneously deficient of the proapoptotic gene *Bax* [26]. The resulting double-KO retinae showed only modestly reduced RGC precursors, suggesting that *ATOH7*-associated absence of RGCs is ultimately caused by accelerated cell death rather than a lack of fate specification. Furthermore, double-KO RGCs displayed grossly normal expression profiles, fired action potentials, and integrated into the retinal circuitry. However, these *Atoh7*-deficient retinae exhibited strong axonal guidance defects of surviving RGCs, as well as the typical vascular abnormalities, concluding that both of these developmental defects are directly related to *Atoh7*-dependent RGCs.

During the duration of our present study, two independent groups have identified downstream targets of ATOH7 in mouse retinae using high-throughput chromatin immunoprecipitation methods [26,27]. Utilizing Cleavage Under Targets and Release Using Nuclease with sequencing (CUT&RUN-seq), Brodie-Kommit et al. observed that ATOH7 binds and induces RGC-associated genes while simultaneously regulating neurogenic targets, such as Notch-associated effectors, as well as inhibiting PR fate-specification genes. Last year, Ge et al. investigated the epigenetic landscape of retinal cell differentiation using single-cell Assay for Transposase-Accessible Chromatin with sequencing (scATAC-seq) integrated with Cleavage Under Target and Tagmentation with sequencing (CUT&Tag-seq) targeting ATOH7. The group similarly described the repression of early naïve RPC (nRPC)-specific genes, the activation of transient RPC (tRPC)-specific transcription factors, and the inhibition of the neurogenic state by regulating the Notch pathway in order to promote differentiation and inhibit pluripotency [27]. Furthermore, the authors described cross-repression between ATOH7 and photoreceptor-specific transcription factors co-expressed in the *ATOH7-positive* tRPCs, the reciprocal inhibition of their targets, as well as the co-activation of targets downstream of the well-defined RGC-determining factors POU4F2 and ISL1 [28].

Our research characterizes for the first time the molecular role of *ATOH7* in human retinal development by identifying highly regulated downstream genes in human induced pluripotent stem cell (iPSC)-derived retinal organoids. Combining CUT&RUN-seq, mRNA sequencing (mRNA-seq) of wildtype (*ATOH7^WT-GFP^*) and mutant (*ATOH7^MUT-GFP^*) *ATOH7* reporter cells, and fixed single-cell RNA profiling (scRNA-seq) of whole wildtype (*ATOH7^WT^*) and KO (*ATOH7^KO^*) organoids (Figure 1), we are able to show that the human organoid model replicates findings in the mouse retinae, while simultaneously providing new candidate genes involved in *ATOH7*-dependent cell fate regulation and putative risk genes that may be involved in pathogenesis. As expected, scRNA-seq revealed a decrease in RGC populations, which was especially accentuated in later stages, and a compensatory increase in cell numbers of primarily cone PRs but also other cell fates. Combining these data with the mRNA-seq of *ATOH7* reporter cells and CUT&RUN-seq targeting ATOH7, we further detected a general ATOH7-dependent activation of RGC genes with a high enrichment of regulated target genes associated with axon development. We also identified the regulation of the progenitor state by the direct regulation of Notch signaling, as well as the inhibition of several cell fate-specific genes. Thus, this study indicates the central and versatile role of ATOH7 in early human neuroretinal development and provides several novel insights into candidates for the potential future therapy development of various RGC-related and vascular eye diseases.

## 2. Materials and Methods

### 2.1. Human iPSC Maintenance

The fibroblast-derived human iPSC line UCSD167i-99-1 was purchased from the WiCell Research Institute (WiCell, Madison, WI, USA). Cells were grown as colonies on truncated recombinant human vitronectin (rhVTN-N; Thermo Fisher Scientific, Waltham, MA, USA) in TeSR™-E8™ culture medium (Stemcell Technologies, Vancouver, BC, Canada) at 36.5 °C and 5.0% CO_2_. Medium changes were performed according to the manufacturer’s weekend-free protocol. Cells were passaged routinely 2–3 times per week at ratios of 1:4–1:12 using Versene Solution (Thermo Fisher Scientific, Waltham, MA, USA). Cell line clones were cryopreserved in the Bambanker™ Standard (Nippon Genetics Europe GmbH, Düren, Germany).

### 2.2. CRISPR/Cas9 Gene Editing

Plasmid design: We designed two single-guide RNAs (sgRNAs) flanking the ATOH7 protein-coding sequence, resulting in an approximately 500 bp deletion (Appendix A). Guide sequences and the top 10 predicted off-targets are presented in Appendix A. Guides were de novo synthesized (Microsynth AG, Balgach, Switzerland) and cloned into an EF1α-PX459 vector, a modified pSpCas9(BB)-2A-Puro (PX459) V2.0 vector (Addgene, Watertown, MA, USA) where the chicken β-actin promoter has been replaced by the EF-1α promoter. ATOH7 eGFP reporter iPSCs were created through homologous recombination (HR) using a plasmid donor. The plasmid design included an ATOH7-P2A-eGFP expression cassette (WT-GFP) or eGFP (MUT-GFP) flanked by 600 bp homology arms, including protospacer adjacent motif (PAM)-site mutations as well as cleavage sites for plasmid linearization (Appendix A). The donor sequence was de novo synthesized commercially and inserted into a pUC57-mini vector backbone (GenScript Biotech, Piscataway, NJ, USA). The BCL2L1_pLX307 vector (Addgene, Watertown, MA, USA) was transiently co-expressed during CRISPR/Cas9 editing to favor the survival of edited cells, as previously described by Li et al. [29].

Gene editing: Human iPSCs were passaged as single cells by 0.75X TrypLE™ Express Enzyme (Thermo Fisher Scientific, Waltham, MA, USA) diluted in phosphate-buffered saline and were seeded at a cell density of 0.3M cells/well in 12-well plates. Reverse transfection was performed by Lipofectamine™ Stem (Thermo Fisher Scientific, Waltham, MA, USA) using 1 μg of total DNA in the proportions of EF1α-PX459:pUC57-mini:BCL2L1_pLX307 2:2:1. The transfected cultures were supplemented with 1X RevitaCell™ Supplement (Thermo Fisher Scientific, Waltham, MA, USA) for 24 h and thereafter with 1 μg/mL puromycin (Sigma Aldrich, St. Louis, MO, USA) for 3 days. Once recovered, cells colonies were dissociated by 0.75X TrypLE™ Express Enzyme and individual cells were sorted by a FACSAria III 4L (BD Biosciences, Franklin Lakes, NJ, USA) into 96-well plates coated with Corning^®^ Matrigel^®^ Matrix hESC-Qualified (Corning, New York, NY, USA), according to the manufacturer’s instructions. The cells were expanded for 10–12 days in StemFlex™ Medium (Thermo Fisher Scientific, Waltham, MA, USA), supplemented with 1X RevitaCell™ Supplement (Thermo Fisher Scientific, Waltham, MA, USA) for the first 48 h.

PCR colony screening: A PCR flanking the ATOH7 CDS was designed to screen CRISPR/Cas9-edited human iPSCs using the primers ATOH7_scr_F2 and ATOH7_scr_R2 (Appendix A). The PCR was performed in 20 μL of the total volume containing 1U TaKaRa LA Taq^®^ DNA Polymerase (Takara Bio Inc., Kusatsu, Japan), 0.4 mM dNTPs (Takara Bio Inc.), 1X LA PCR Buffer ll Mg^2+^ plus (Takara Bio Inc.), 1X Solution S (Solis Biodyne, Tartu, Estonia), and 0.4 μM of the primers. The cycling conditions were as follows: 1 cycle of 94 °C for 2 min; 1 cycle of 98 °C for 10 sec; 35 cycles of [68 °C for 4 min, 72 °C for 10 min]; hold at 10 °C. The PCR products were visualized on a DNA gel (1% agarose) and homozygous clones were further screened by Sanger sequencing using the sequencing primers ATOH7_Ex1_F2 and ATOH7_Ex1_R (Appendix A). For selected human iPSC clones, off-target PCRs were carried out in order to screen four of the top-ranked off-target sites for each CRISPR/Cas9 sgRNA using the primer pairs OT5_1-4 and OT3_1-4 (Appendix A).

Sanger sequencing: The PCR products were pretreated with ExoSAP-IT™ PCR Product Cleanup Reagent (Thermo Fisher Scientific, Waltham, MA, USA), and Sanger sequencing thermocycling was performed using BigDye™ Terminator v1.1 Cycle Sequencing Kit (Thermo Fisher Scientific, Waltham, MA, USA), according to the manufacturer’s instructions. Reactions were cleaned by ethanol precipitation and were sequenced on a SeqStudio™ Genetic Analyzer version 1.1 (Thermo Fisher Scientific, Waltham, MA, USA).

### 2.3. Retinal Organoid Differentiation

Once human iPSCs reached approximately 60–70% confluency, retinal differentiation was performed using the 2D/3D differentiation protocol described in detail by Gonzalez-Cordero et al. and Cuevas et al. [30,31]. In short, cells were cultured in Essential 6™ Medium (E6; Thermo Fisher Scientific, Waltham, MA, USA) for two days, followed by neural induction medium (NIM). During differentiation week 4, organoids were dissected by 19G needles attached to a syringe and transferred to a suspension culture in retinal differentiation medium (RDM). The culture medium was switched to fetal bovine serum- (FBS; Thermo Fisher Scientific, Waltham, MA, USA) and taurine-containing (Sigma Aldrich, St. Louis, MO, USA) retinal differentiation medium 1 (RDM1) from differentiation week 6. All organoid cultures were supplemented with 1X antibiotic–antimycotic (Thermo Fisher Scientific, Waltham, MA, USA).

### 2.4. Quantitative Reverse-Transcription PCR

The total RNA was extracted using NucleoSpin^®^ RNA (Macherey-Nagel, Düren, Germany) according to manufacturer instructions. Reverse transcription was performed using Superscript™ III First-Strand Synthesis SuperMix (Thermo Fisher Scientific, Waltham, MA, USA) and oligo(dT)_20_ primers. The synthesized cDNA was diluted to a concentration equivalent to 10 ng/μL of the input RNA.

Quantitative reverse transcription PCR (RT-qPCR) was performed using SYBR^®^ Select Master Mix (Thermo Fisher Scientific, Waltham, MA, USA) in 20 μL of the total volume containing 400 nM of RT-qPCR primers (Appendix A) and 1 μL of cDNA. Thermocycling was performed on the LightCycler^®^ 480 Instrument II (F. Hoffmann-La Roche AG, Basel, Switzerland) using a 2-step cycling program [1 cycle of 50 °C for 2 min; 1 cycle of 95 °C for 2 min; 45 cycles of 98 °C for 15 s and 60 °C for 1 min].

Ct values were determined by second derivative maximum analysis using the LightCycler^®^ 480 Software version 1.5.1.62 SP3 (F. Hoffmann-La Roche AG, Basel, Switzerland). Technical triplicates were averaged and normalized to the mean GAPDH expression and calibrated to the normalized expression in human iPSCs, according to the 2^−ΔΔCt^ method [32]. GraphPad PRISM v6.07 (GraphPad Software, San Diego, CA, USA) was used for all statistical calculations and visualizations. The amplified products were confirmed by product melting temperature analysis and by DNA gel electrophoresis.

### 2.5. Immunocytochemistry and Axon Sprouting Assay

Retinal organoids were fixed in 4% paraformaldehyde (PFA)/PBS for 15 min at room temperature and thereafter preserved in 30% sucrose/PBS at 4 °C overnight. The prepared retinal organoids were frozen in tissue freezing medium (BioSystems, Muttenz, Switzerland), cut in 12 µm thick sections, and stored on SuperFrost^®^ Plus adhesion slides (Thermo Fisher Scientific, Waltham, MA, USA) at −80 °C until immunostaining. The slides were washed 3 × 5 min with IHC wash buffer (20 mM pH 7.6 Tris-HCL, 14 mM NaCl, 0.05% *w*/*v* Triton X-100) between each incubation step. The samples were blocked in IHC blocking buffer 1 (20 mM pH 7.6 Tris-HCL, 14 mM NaCl, 0.2% *w*/*v* Triton X-100, 3% *w*/*v* BSA and 5% Donkey serum [Sigma-Aldrich, St. Louis, MO, USA]) for 1 h at room temperature. Primary antibodies were diluted in IHC blocking buffer 1 and added on top of the organoid sections demarked by a hydrophobic PAP pen and incubated at 4 °C overnight in a humidified chamber (Appendix A). Secondary antibody incubation was performed in IHC blocking buffer 2 (20 mM pH 7.6 Tris-HCL, 14 mM NaCl, 2% *w*/*v* BSA, 0.2% *w*/*v* Triton X-100) for 2 h at room temperature in a humidified chamber (Appendix A). Cover slips were mounted using the Flouromount-G^®^ mounting medium (SouthernBiotech, Birmingham, AL, USA) containing DAPI. Imaging was conducted by an SP8 Confocal Laser Scanning Microscope (Leica Microsystems CMS GmbH, Wetzlar, Germany) with a 63x HC PL APO CS2 (NA 1.40) oil immersion objective. Laser sources were set to 405, 488, and 633 nm for DAPI, green, and red channels, respectively. Images were acquired and exported using the Leica Application Suite X version 3.7.1 (Leica Microsystems CMS GmbH, Wetzlar, Germany).

To study axon sprouting, retinal organoids generated from 4 differentiations and at least 12 organoids per condition were cultured on 22 × 22 mm coverslips coated with Corning^®^ Matrigel^®^ Matrix for Organoid Culture (Corning, New York, NY, USA). Prior to incubation, the coverslips were sterilized by 70% ethanol, dispensed into 6-well plates, and preincubated with a thin layer of undiluted Matrigel^®^ for 2 h at 37 °C. Week 7 reporter organoids were placed on top of the coverslip and incubated in RDM1 with regular medium changes. Alternatively, Matrigel^®^ droplets containing reporter organoids were made by submerging an organoid in 100 μL of undiluted Matrigel and thereafter dispensing the entire volume as a droplet inside a 6 cm dish, using a wide-bore pipette tip. Droplets were incubated for 30 min in 37 °C and were thereafter cultured in RDM1 with regular medium changes.

### 2.6. Retinal Cell Dissociation

Entire week 7 organoids were dissociated for CUT&RUN-seq, *ATOH7* reporter mRNA-seq, and scRNA-seq experiments using the Worthington Papain Dissociation System (Worthington Biochemical Corporation, Lakewood, NJ, USA). Approximately 40 week 7 organoids were collected per dissociation for reporter-enriched mRNA-seq and 10 for PFA-fixed scRNA-seq, requiring approximately 1:10 of the recommended reagent volumes specified in the manufacturer’s dissociation protocol. The organoids were incubated in a Papain/DNase mix at 37 °C for 35 min under constant agitation following the addition of DNase and ovomucoid solution, and gentle trituration using a p1000 pipette. Debris was reduced using the optional discontinuous density gradient step and cells were resuspended in RDM1 and finally passed through a 40 µm strainer. For CUT&RUN-seq and *ATOH7* reporter mRNA-seq, cell suspensions were sorted by FACS using a FACSAria III 4L (BD Biosciences, Franklin Lakes, NJ, USA).

### 2.7. Cut under Target and Release Using Nuclease Sequencing

Sample preparation: Dissociated cells from week 7 *ATOH7^WT-GFP^* organoids were resuspended in RDM1 and fixed for 2 min by the addition of 0.1% PFA at room temperature. Cross-linking was arrested by the addition of 10% *v*/*v* 1.25 M of glycine, and the cells were centrifuged at 1000 G for 3 min at 4 °C and finally resuspended in RDM1. Cells were subsequently sorted using FACS according to eGFP-signal and were thereafter prepared according to the manufacturer’s instructions using a CUT&RUN Assay Kit (Cell Signaling Technology, Danvers, MA, USA). A total of 1.5 × 10^5^ cells were used per reaction. The enriched DNA was prepared for sequencing using the NEBNext^®^ Ultra™ II DNA Library Prep Kit (New England Biolabs, Ipswich, MA, USA). Due to low input amounts, the libraries were amplified by PCR (15 cycles). Paired-end 50 bp short-read sequencing was performed on Illumina NextSeq2000 (Illumina, San Diego, CA, USA).

Raw data and mapping: Sequencing data were processed either through the Galaxy Europe platform (http://usegalaxy.eu accessed on 29 March 2024) or in local environments using Conda version 23.1.0 (Anaconda, Austin, TX, USA). Reads were assessed by FASTQscreen v0.14.0 and Kraken2 v2.1.1 for contaminants and FASTQC v0.12.1 for read quality. The trimming of the reads was performed with cutadapt v4.2 with the settings of -a AGATCGGAAGAGCACACGTCTGAACTCCAGTCA \--quality-cutoff 20,20 \--trim-n. Read mapping was performed against the reference genome RefSeq GRCh38p14 and output as BAM files using Bowtie2 v2.5.0 (Galaxy) with the parameters of -I 10 \-X 700 \--no-mixed \--no-discordant \--local. The mappings were filtered by removing duplicate reads using Samtools fixmate v. 1.15.1 and markdup v1.15.1 (Galaxy), and by removing reads with a mapping quality of MAPQ < 30 using Samtools Filter SAM or BAM v2.1.1 (Galaxy). Generated BAM files were merged using Samtools merge v1.15.1 (Galaxy). Coverage was analyzed by converting BAM files to the bigWig format with a 25 bp bin size using bamCoverage v3.5.1.0.0 (Galaxy) and was visualized in IGV v2.15.4. Heatmaps showing the enrichment of reads within 5kb of known TSSs were constructed through bedtools v2.30.0 with commands computeMatrix and plotHeatmap, using bigWig files as input.

Peak calling and P2G: Peak calling was performed with MACS2 callpeak v2.2.7.1 (Galaxy) using merged eGFP_IgG as the control with the parameters of --format Paired-end BAM \--gsize 3137300923 \--*p* value 1 × 10^−5^. Consensus peaks between replicates were defined by IDR using IDR v2.0.3 (Galaxy) with the parameter of --soft-idr-threshold 0.05. Blacklisted regions described by Nordin et al. were removed from the output [33]. For motif discovery, a BED file containing 200 bp regions centered around summits of the consensus peaks was used as input for HOMER findMotifsGenome v4.11 (Galaxy) using the parameter of -size 200. Consensus peaks were analyzed in RStudio v2022.12.0 + 353 running R v4.2.2. Genomic element distribution was assessed by Chipseeker v1.34.1. P2G analysis was performed using ChIP-Enrich v2.22.0 using the variables of locusdef = “enhancer_plus5kb”, method = “chipenrich”.

### 2.8. Reporter-Enriched mRNA Sequencing

Sample preparation: A total of four independent differentiations were performed per condition. Total RNA was extracted from 0.5 M sorted cells per sample using NucleoSpin^®^ RNA (Macherey-Nagel, Düren, Germany) according to manufacturer instructions. Sequencing libraries were prepared by Truseq Stranded mRNA (Illumina, San Diego, CA, USA) protocol and sequenced as 100 bp single-end reads on a Novaseq 6000 system (Illumina, San Diego, CA, USA).

Raw data and mapping: Sequencing data were processed in local environments using Conda 23.1.0. The reads were assessed by FASTQscreen v0.14.0 and Kraken2 v2.1.1 for contaminants and FASTQC v0.12.1 for read quality. The trimming of reads was performed with cutadapt v4.2 with the settings of -a AGATCGGAAGAGCACACGTCTGAACTCCAGTCA \--minimum-length 30 \--quality-cutoff 20,20 \--trim-n. Pseudoalignment was performed with the salmon v1.10.0 quant function with the settings of --fldMean 150 \--fldSD 70 \--validateMappings, using concatenated RefSeq GRCh38p14 DNA and RNA assemblies (gentrome) as the index and the corresponding RefSeq GRCh38p14 DNA assembly as the decoy. Conventional alignments for the visualization of the mRNA expression within the ATOH7 locus were performed using the RNA STAR version 2.7.11 (Galaxy) and BWA-MEM2 version 2.2.1 (Galaxy) with ATOH7 reporter sequences as the reference.

Differential expression analysis: All downstream analysis was carried out in RStudio v2022.12.0+353 running R v4.2.2. Genes with a low expression were filtered out using edgeR:: filterByExpression v3.40.2, and the differential expression analysis was performed by DESeq2 v1.38.3. The correction of fold changes was performed with DESeq2::lfcShrink using ashr v2.2-54. Genes were considered differentially expressed if Benjamini–Hochberg FDR < 0.01. The enrichment analysis was performed with WebgestaltR v0.4.5 using the parameters of referenceSet = “genome_protein-coding”, minNum = 30, maxNum = 500.

### 2.9. Gene Enrichment Analysis

Enrichment was carried out by g:Profiler (https://biit.cs.ut.ee/gprofiler/gost, accessed on 15 October 2023) using g:SCS threshold of > 0.05 for multiple testing on annotated genes, and the results were filtered for terms comprising 30–600 genes. GO enrichment results were further reduced using REVIGO (http://revigo.irb.hr, accessed on 17 October 2023). Biological pathway enrichment was performed both on annotated as well as all genes using terms from the KEGG, REACTOME, and WikiPathways. Disease-related terms were filtered out.

### 2.10. Fixed Single-Cell RNA Sequencing

To avoid any potential composition bias due to the cell death of sensitive retinal neurons, we opted to performed PFA-fixed and hybridization-based scRNA-seq using the Chromium Fixed RNA Profiling technique (10X Genomics, Pleasanton, CA, USA), according to the manufacturer’s instructions. Approximately 10 week 7 organoids per sample (*ATOH7^W^*^T^ and *ATOH7^KO^*) were dissociated by Papain according to the above-mentioned methodology, resulting in at least 1M cells with a viability of > 85% before fixation. Sequencing was performed on a Novaseq 6000 (Illumina, San Diego, CA, USA) as paired-end reads. Read mapping and count quantification were performed with Cell Ranger multi v7.0.0 targeting 18’075 genes with GENCODE GRCh38p13 as the reference. Count matrices were uploaded to the Cellenics^®^ platform (Biomage Ltd., Edinburgh, UK), and low-quality cells were further filtered on a minimum number of unique molecular identifiers (UMIs) per cell > 2000, mitochondrial fraction < 10%, linear gene-to-UMI count correlation with a *p*-value < 0.001, doublet-scoring, and a scDblFinder doublet probability threshold of 0.4. Samples were integrated using Harmony [34], using log normalization on 2000 highly variable genes, with 28 principal components explaining > 90% of the variation within the dataset. Embedding was visualized using UMAP and cell clustering was performed with the Louvain clustering method. Trajectory inference was computed with Monocle3.

## 3. Results

### 3.1. CRISPR/Cas9 Gene Editing of Human iPSCs

We created *ATOH7^KO^* (Figure 2) human iPSCs by the CRISPR/Cas9-mediated deletion of the entire *ATOH7* coding sequence (*ATOH7* CDS) by non-homologous end joining (NHEJ) using dual single-guide RNAs, and during the editing procedure, we also isolated isogenic, unedited iPSCs (*ATOH7^WT^*; Figure 2). Using the same approach, we also created WT *ATOH7* reporter lines (*ATOH7^WT-GFP^*, Figure 2) where we inserted an *ATOH7-P2A-eGFP* sequence into the excised region (*ATOH7* CDS) by CRISPR/Cas9 plasmid-mediated HR (Appendix A). The P2A sequence allows the separation of the reporter polypeptide by ribosomal skipping during translation and is highly efficient in human cells [35]. Conversely, we also attempted to create mutant ATOH7 reporter (*ATOH7^MUT-GFP^*, Figure 2) lines by plasmid-mediated HR, where the donor sequence inserted an eGFP sequence at the target site (500 bp *ATOH7* CDS deletion) (Appendix A). Target and predicted off-target sites were verified by Sanger sequencing in all edited iPSC clones (Appendix A). Upon the inspection of mRNA sequences of differentiated *ATOH7^MUT-GFP^* organoids, we detected additional sequences in the *ATOH7* target locus consisting of an inversion of the cut *ATOH7* CDS, as well as parts of the homology arms of the plasmid donor (Appendix A). No WT *ATOH7* CDS was detected in the mRNA-seq of *ATOH7^MUT-GFP^* organoids.

### 3.2. Mutant Human iPSC-Derived Organoids Recapitulate Axon Deficiency

*ATOH7^WT^* and *ATOH7^KO^* iPSCs were differentiated into early-stage retinal organoids by an established feeder-free 2D/3D differentiation protocol [30,31]. We detected ATOH7-positive cells by immunofluorescence as early as differentiation week 6, with high amounts of expressing cells during weeks 7–8, after which expression steadily declined, and thus we chose to study the role of *ATOH7* primarily in week 7 differentiations (Figure 3A). *ATOH7^WT^* organoids showed an organized inner layer of cells immunoreactive for the RGC markers POU4F1/2 and RBFOX3, while *ATOH7^KO^* organoids did not show any immunoreactivity towards ATOH7 and showed low amounts of less organized, presumed RGCs (Figure 3B). Furthermore, it appeared that RCVRN/CRX-positive PR precursors were more numerous in the *ATOH7^KO^* organoids and began to organize a layer on the outer side of the neuroretinal rim (Figure 3B).

Retinal organoids generated from *ATOH7^WT-GFP^* and *ATOH7^MUT-GFP^* iPSCs showed a similar macroscopic morphology in brightfield and an obvious eGFP signal, which was more pronounced towards the center of the neuroretinal rim (Figure 3C). We observed no *ATOH7* mRNA expression in different batches of *ATOH7^MUT-GFP^* (Figure 3D). In contrast to *ATOH7^MUT-GFP^* organoids, the incubation of *ATOH7^WT-GFP^* week 7 organoids on Matrigel^®^-coated coverslips resulted in the growth of numerous projections, which were identified as axons by immunoreactivity to α-TUBB3 and α-TAU antibodies (Figure 3E). The generation of projections was further enhanced by cultivating retinal organoids encapsulated in droplets of concentrated Matrigel^®^, where also *ATOH7^MUT-GFP^* organoids were seen to generate very few sprouts (Figure 3F).

### 3.3. The Majority of ATOH7 Targets Are Distal Enhancer Sites

*ATOH7^WT-GFP^* organoids were dissociated, enriched by Fluorescence-Activated Cell Sorting (FACS), and subjected to CUT&RUN-seq (Appendix A) [36]. Assays were performed using eGFP-positive cells treated by anti-ATOH7 antibodies (target), eGFP-negative cells treated by anti-ATOH7 antibodies (negative control), and eGFP-positive cells treated by non-immune IgG (background control) (Appendix A). Enriched DNA libraries were amplified and sequenced as 50 bp paired-end reads on an Illumina NextSeq 2000.

Reads were mapped to RefSeq GRCh38p14 using Bowtie2, and alignments were subsequently visualized as binned sequencing coverages [37]. Target replicates displayed the enrichment of sequences around known transcription start sites (TSSs), in contrast to negative and background controls (Figure 4A). Consensus peaks of replicate experiments for target and negative control were identified by the irreproducible discovery rate (IDR) testing of peaks generated by MACS2 analysis against the background control (Figure 4B) [38,39]. A total of 1379 target consensus peaks were identified (Appendix A), whereas the analysis of the negative control peaks generated 38 consensus peaks, with 1 overlapping peak between the two sets (Figure 4C). Approximately 16% of the identified peaks were found within ±1 kb of known TSSs and, in total, 22% were found within ±5 kb of TSSs. In contrast, a majority of peaks were likely remote enhancers, most of which were found in distal intergenic regions (Figure 4D).

### 3.4. Top Annotated Genes Are Involved in RGC Differentiation, Axonogenesis, and Notch Signaling

Considering a majority of target consensus peaks were located outside of known promoter regions, we utilized ChIP-Enrich for peak-to-gene (P2G) annotations (Appendix A), since this tool utilizes known enhancer locations as well as in-silico predictions based on the chromatin structure to enhance the P2G assignment of distal enhancers [40]. A total of 1644 unique target genes were identified through P2G, as some peaks were annotated to more than one target gene. We saw P2G annotations of several RGC regulatory, structural, and functional genes, which were identified as ATOH7 targets in previous studies on mouse retinae, including *DCC*, *DCX*, *EBF2*, *ELAVL4*, *EYA2*, *GAP43*, *IGFBLP1*, *ISL1*, *ISLR2*, *KLF7*, *NHLH1*, *PBX3*, *POU4F2*, *POU6F2*, *SATB1*, *SOX11*, and *TLE4* [26,27]. There was 9.0% overlap (265 genes) in uniquely annotated human orthologous genes between our dataset and CUT&TAG data from Ge et al. and 12% overlap (442 genes) between our dataset and CUT&RUN data from Brodie-Kommit et al., in comparison to 14% overlap (483 genes) between data from the two previous authors (Appendix A) [26,27]. This indicates that, despite different techniques and species, a comparable low level of consensus is seen in target gene identification between different studies, thus warranting more replicative studies. In total, 34% (555 genes) of ATOH7 target genes identified in our human retinal organoid data were replicated in at least one of the aforementioned mouse retinae studies.

Several genes were annotated by multiple peaks in our dataset, with the sulfotransferase *HS6ST1* being annotated the most (Figure 4E and Appendix A). Intriguingly, *Hs6st1* has been shown to be crucial for retinal axon guidance in mice [41,42]. Amongst the genes annotated to the most significant peaks ranked by IDR were the well-known RGC genes and well-established ATOH7 targets *POU4F2*, *ISL1*, and *EYA2*, as well as *ATOH7* itself, represented by a peak targeting the known and disease-associated upstream distal enhancer element (Figure 4E and Appendix A) [11,43]. Amongst the highly significant genes were also several axon guidance effectors, such as *CNTN2, EFNA4*, *EFNA5*, *EPHA5*, *NECTIN1*, and *TUBB3*, as well as the Notch pathway-associated genes *HDAC2*, *HDAC4, NCOR2, TLE1*, *TLE2*, and *TLE4.* Gene ontology (GO) enrichment using g:Profiler discovered terms related to axonogenesis, eye development, and Wnt signaling pathway amongst the top biological processes (Figure 4F) [44].

### 3.5. Motif Analysis Reveals Enrichment of Loci Bound by RPC, RGC, and PR-Associated Factors

The Hypergeometric Optimization of Motif Enrichment (HOMER) analysis discovered a total of 242 significantly enriched known motifs amongst the target consensus peaks, with the top-ranked motif being the bHLH transcription factor-associated E-box modeled by Atoh1 (Table 1, Appendix A) [45]. The de novo motif analysis identified 25 enriched sequences, of which 14 showed a significant match to known transcription factor binding sites, of which the most significantly enriched sequence matched bHLH transcription factor motifs (Table 2, Appendix A).

We also saw the enrichment of a TAATT core sequence, matching motifs bound by homeobox factors, with closest similarity to EN1 and DLX2 binding sites. The enrichment of EN1 binding sites within ATOH7 target consensus peaks corresponds well to the P2G annotation of several ephrin guidance effectors, as EN1 is known to regulate the expression of these in the central visual centers [46]. DLX1/2 are crucial for the survival of RGCs and have been proposed to function both downstream of ATOH7, as well as in parallel and cooperatively, in order to regulate *Pou4f2* and RGC differentiation [47,48]. The same consensus sequence also displayed a high similarity to the RPC transcription factors LHX1 and LHX2 [49,50]. A further enriched sequence matched the target motif of MEIS2, an atypical homeodomain-containing gene that, together with MEIS1, redundantly promotes retinal neurogenesis by positively regulating the expression of RPC genes, Notch effectors, and bHLH factors, including *ATOH7* [51].

Additionally, a GGCGG core sequence was identified, which was best predicted to be bound by SP2, a zinc finger transcription factor regulating vitally important cellular functions, including cellular proliferation. However, other SP family members also bind the GGCGG core sequences, and in particular, SP4 is expressed in retinal neurons where it is strongly implicated in PR development, as it activates both rod- and cone-specific genes and promotor sites in synergy with the cone–rod homeobox transcription factor CRX [52]. Finally, amongst the highly significant binding sequences was a motif matching DUX4, albeit with a relatively low sequence similarity score. This homeobox transcription factor is expressed in embryonic tissue, promoting, amongst others, cell proliferation [53].

### 3.6. Differentially Expressed Genes in Reporter Cells Reflect ATOH7 Target Gene Categories

Next, we sought to confirm our findings at a transcriptional level in *ATOH7* lineage cells. Early-stage organoids derived from *ATOH7^MUT-GFP^* and *ATOH7^WT-GFP^* iPSC lines were dissociated, eGFP-expressing cells were enriched by FACS (Appendix A), and total mRNA was amplified and sequenced as 100 bp single-end reads on an Illumina instrument. Sequenced reads were pseudoaligned to RefSeq GRCh38p14 RNA assembly and transcripts were quantified by Salmon (Appendix A) [54]. A differential expression analysis was performed by DESeq2 using a significance cutoff for adjusted *p*-value > 0.01, which was based on the Benjamini–Hochberg false discovery rate (FDR) [55]. A total of 4037 (2124 down, 1913 up relative to *ATOH7^MUT-GFP^*) differentially expressed genes (DEGs) were detected in eGFP-enriched cells (Appendix A), of which the top candidates are presented in Figure 5A,B. Amongst the most significantly downregulated DEGs, we found several RGC– and axonogenesis-related genes, including *AFAP1*, *CNTN2*, *DCC*, *DLX1*, *EBF3, EYA2*, *GAP43*, *GOXG1*, *ISL1*, *KLF7*, *POU4F2*, *PRPH*, and *SNCG*, of which several were present amongst *ATOH7* target genes.

In contrast, the most significantly upregulated DEGs included PR-specific genes, such as *ABCA4, AIPL1, CRX, NEUROD1, NR2E3,* and *RPGR* [56,57,58,59,60,61]. Also of notice was the presence of *FOXD1*, a regulator of ipsilateral axon projections in mammals, as well as *VSX1*, a homeobox important in cone bipolar cell development [62,63]. Only a handful of the top upregulated DEGs were present amongst ATOH7 target genes. This includes the neuroretinal type-2 cadherin *CDH7* and the proto-oncogene *MN1*, both of which displayed two enhancer binding sites for ATOH7 [64,65]. The only top upregulated DEG displaying a promoter-associated binding site was kainate receptor *GRIK3*, which is transiently expressed in immature rods and mediates glutamate-dependent signaling from light-sensitive ipRGCs [66].

The GO enrichment of the presented DEGs by g:Profiler resulted in the term “axon development” as the top enriched biological process (Figure 5C), corresponding well to the enrichment of terms “axonogenesis” and “regulation of neuron projection development” amongst ATOH7 target genes (Figure 4F).

### 3.7. Loss of ATOH7 Causes Shift in Cell Type Composition in Human Retinal Organoids

To gain better insights into the identities of retinal cells expressing *ATOH7* and its target genes, we performed scRNA-seq on *ATOH7^WT^* and *ATOH7^KO^* week 7 whole-organoid dissociations using Chromium Fixed RNA Profiling (10X Genomics, Pleasanton, CA, USA). The sequencing yielded 11482 cells from *ATOH7^WT^* organoids and 13339 cells from *ATOH7^KO^* (8687 and 10210 cells after quality filtering, respectively). The analysis was performed on the Cellenics^®^ platform (Biomage Ltd., Edinburgh, UK) using Harmony for sample integration, with 28 principal components explaining >90% of the expression variation [34].

Based on Louvain clustering, six main cell identities could be established. These were highly similar to previously published data on human fetal retinae and retinal organoids and consisted of nRPCs expressing marker genes, such as *SFRP2* and *VSX2;* tRPCs expressing marker genes, such as *ATOH7*, *HES6*, and *GADD45A*; early RGCs expressing marker genes, such as *CNTN2* and *POU4F2*; late RGCs expressing marker genes, such as *NSG1* and *GAP43*; horizontal and amacrine cells (H&As) expressing marker genes, such as *ONECUT2* and *PTF1A*; and PRs expressing marker genes, such as *OTX2* and *CRX* (Figure 6A and Appendix A) [67]. As anticipated, *ATOH7* was mostly expressed in tRPCs and early RGCs; however, *ATOH7*-positive cells were present in all clusters (Figure 6A). Pseudotime inference confirmed that RGC, H&As, as well as PRs all emerged through the common multipotent tRPC (Figure 6B). *ATOH7^KO^* organoids showed a decreased number of RGCs compared to WT ones, with an approximately 50% reduction in early RGC numbers and 80% reduction in late RGCs, corresponding well to the previous observations on *ATOH7*-independent RGC generation of the accelerated RGC loss in later stages of *ATOH7*-deficient retinae (Figure 6C) [22,23,26]. As expected, there was also an increased number of other early cell types, confirming the central role of ATOH7 in regulating multiple cell fates (Figure 6C) [22,23,24,25].

To highlight the direct targets of ATOH7 during neuroretinal development, we re-clustered our scRNA-seq data using only *ATOH7*-positive cells. The new clustering retained all the original cell types of the scRNA-seq data, albeit with shifted cell compositions towards tRPCs and early RGCs (Figure 6D and Appendix A). Next, we integrated identified cell type-specific marker genes in *ATOH7*-positive cells with DEGs in *ATOH7* reporter cells and identified ATOH7 target genes in order to highlight candidate marker genes directly regulated by ATOH7 (Figure 6E, Appendix A). The majority of the targeted marker genes identified through this method were expressed in RGCs, most of which were positively regulated by ATOH7. Herein were included the well-established ATOH7 targets *POU4F2*, *ISL1*, and *ELAVL4*, which displayed several binding sites. The strongest differentially expressed genes were *DCC* and *LDB3*, which together with *AFAP1* were identified as the topmost significantly regulated DEGs (Figure 5A,B). Conversely, targeted marker genes mainly expressed in progenitors were preferentially negatively regulated by ATOH7, in particular the RGC/amacrine fate regulator *FOXN4*, as well as the NOTCH signaling-associated genes *KIAA1217*, *HES6,* and *NOTCH1* [68,69]. ATOH7 also showed a negative regulation of several PR-specific markers, including *PDE6B*, *VXN*, and *BCL2L1*.

### 3.8. Differentially Expressed ATOH7 Target Genes and Pathway Enrichment

To further study the role of ATOH7 on the developing neuroretina, we overlapped ATOH7 target genes with the DEGs identified in *ATOH7* reporter cells, resulting in 469 ATOH7 target DEGs (Figure 7A and Appendix A). In agreement with earlier observations, the vast majority of RGC-specific ATOH7 target DEGs were downregulated in mutants, whereas a majority of genes associated with tRPCs, PRs, and to some extent H&As were upregulated (Figure 7B). Corresponding to previous analyses, the GO enrichment of ATOH7 target DEGs by g:Profiler resulted in “axonogenesis” being the top associated biological process. Additionally, pathway enrichment was performed by analyzing terms annotated by KEGG, REACTOME, and WikiPathways, resulting in “Nervous system development”, “Axon guidance”, and “Notch signaling” as the most significant terms amongst annotated genes, where Notch signaling presented the highest gene enrichment ratio (Figure 7C–E, Appendix A). Considering that *ATOH7* deficiency causes vascular changes and the accelerated death of ATOH7-independent RGCs, we were also interested in identifying potential ATOH7-regulated secreted factors and therefore annotated *ATOH7* reporter DEGs to the GO term GO:0005576 “Extracellular Region” (Figure 8A,B). The analysis revealed a direct and indirect ATOH7 association with the expression of several candidate genes encoding extracellular and secreted factors, including the Wnt signaling genes *WNT10A* and *NOTUM* and the BMP regulator gene *BMPER*.

### 3.9. Expression of PHPV/EVR-Related Gene NDP Is Reduced in ATOH7 Mutants

In addition to *ATOH7*, only a handful of genes have been linked to the pathogenesis of PHPV and FEVR. To date, pathogenic sequence variants in human genes associated with both phenotypes include *ATOH7*, *NDP, FZD4*, and *ZNF408*; genes causative of PHPV include *FOXC1*, *PITX2*, *COX15*, and *PAX6*; and genes causative of EVR include *LRP5*, *TSPAN12*, *KIF11*, *RCBTB1*, and *CTNNB1* [70,71,72,73,74,75,76,77,78,79,80,81,82,83,84,85,86]. Animal studies have discovered additional genes associated with PHPV, which include orthologs of *EFNB2, EFNA5, SKI, BAX, BAK1, FZD5, ANGPT2, TP53, CDKNA2, CRYBA1, TGFB2*, *VEGFA*, and *NEO1* [87,88,89,90]. Finally, *KDR, MYCN, PAX2,* and *SMAD2* have been identified as PHPV-associated candidate genes through bioinformatic approaches [87].

IPSC-derived retinal organoids do not contain a vascular system. Consequently, the vast majority of PHPV/EVR-related genes are unlikely to be expressed in this model. To answer this question in more detail, we filtered *ATOH7* reporter DEGs to only include PHPV and/or the EVR-associated genes described above and included integrated data from all sequencing experiments (Figure 8C). Within this gene set, *EFNA5* was the only gene directly regulated by ATOH7. However, it was of utmost interest that the Norrie disease-causing gene *NDP* showed the strongest downregulation in mutant reporter cells. This is likely a secondary effect of the RGC loss, as *NDP* was mostly expressed in RPCs, and no associated ATOH7 binding was detected to the regulatory regions of *NDP*. Interestingly, expression could also be seen in tRPC and early RGC clusters, where the vast majority of *NDP*-expressing tRPCs and all RGCs were *ATOH7*-positive (Figure 8D). Furthermore, the well-characterized NDP-receptor *LRP5* showed an upregulation in mutant reporter cells, with a visible increase in tRPCs and early RGCs as a possible compensatory mechanism for reduced Wnt signaling by NDP (Appendix A).

## 4. Discussion

To our knowledge, two previously published studies have identified downstream ATOH7 targets in the mouse retina using CUT&RUN-seq and CUT&Tag-seq, respectively [26,27]. Our study contributes to this body of evidence by confirming several previously observed targets as well as offering novel candidates in human neuroretinal development (Appendix A). The rather low congruence in ATOH7 target genes between these studies may reflect differences in the biological model, method, and design of target DNA enrichment and analysis, but may also reflect temporal differences in target binding during neuroretinal development. Furthermore, the de novo motif discovery of enriched DNA sequences corresponded well to the earlier observation by Ge et al. that ATOH7 may compete for binding regions targeted by DLX1/2, ISL1, as well as members of the ONECUT and SOX families (Appendix A) [27].

However, we observed neither the enrichment of the CRX/OTX motif amongst enriched binding regions nor the ATOH7 binding of regulatory elements associated with these genes, unlike those described in mouse retinae studies [26,27]. Instead, we observed the enrichment of a GGCGG core sequence that could confer binding by the SP zinc finger transcription factor family members. The neuron-specific SP4 activates both the *RHO* and *PDE6B* promoters and is abundantly expressed in all retinal layers [52]. Additionally, SP1 activates the *RHO* promoter, and both SP1 and SP3 competitively repress the SP4-mediated activation of *PDE6B*. All three homologs show immunoprecipitation with CRX in vitro and functional synergy with CRX at the *RHO* promoter. An overlap between ATOH7 and SP family gene targets could thus be an alternative RGC/PR fate regulation mechanism to the proposed co-regulation with CRX/OTX.

### 4.1. ATOH7 Is Directly Involved in Axon Development

Considering that both our study as well as previous ones have identified a large number of potential ATOH7 target genes, we chose to focus on a subset of relevant genes that showed significant differential expression in enriched reporter cells and simultaneously possessed the binding of ATOH7 to regulatory elements (promoter or enhancer regions) as indicated by CUT&RUN sequencing (Figure 7D, Appendix A). Both GO and biological pathway analyses within this gene set indicated genes related to axon development and guidance to be the most significantly enriched. Many of these genes were also identified as RGC-specific cell markers (Figure 6E). The observation that ATOH7 directly controls genes responsible for maintaining the identity of RGCs agrees with the observation that a significant number of ATOH7-targeted genes overlap with gene targets of RGC-specifying neurogenic transcription factors, such as POU4F2 and ISL1 [27].

*DCC,* which was found to have an enhancer bound by ATOH7, may very well explain the strong axonal deficiency caused by *ATOH7* mutations, as the deletion of this netrin-1 receptor causes ONH in mice [91]. This gene has been previously associated with *Atoh7* expression and was amongst the most significantly downregulated genes in mutant reporter cells in the present study (Figure 5B) [92]. Also, *CNTN2* was amongst the strongly downregulated genes and presented ATOH7 binding its promoter in addition to several enhancer-binding sites. The *CNTN2*-encoded neural cell adhesion molecule is established in neuronal migration and axon fasciculation, and, notably, KO RGCs lose their ability to extend axons on Cntn2-coated surfaces [93].

Several of the enriched axon guidance effectors could contribute not only to the associated defect in optic nerve development, but also to the vascular aberrations that PHPV and FEVR entail. Guidance cues expressed on the axons of *ATOH7*-positive RGCs could directly interact with the hyaloid at and beyond the optic disc, as well as with the forming retinal vasculature and associated astrocytes at the retinal nerve fiber and RGC layers [94]. For example, the ATOH7 target DEG *SEMA6A* encodes a transmembrane semaphorin ligand, which regulates the survival and growth of endothelial cells by controlling the expression and signaling of the VEGF receptor VEGFR2. *Sema6a*-mutant mice exhibit defects in hyaloid complexity, as well as an abnormally reduced extension of the vascular network between the optic nerve and periphery [95]. The reduced extension was not seen in older mice, suggesting that SEMA6A is a time-limited contributor to retinal vessel development.

Another highly relevant axon guidance gene found amongst ATOH7 target DEGs was *EFNA5*, which was associated with several possible ATOH7 enhancer sites. This ephrin ligand has been shown to mediate repulsive axon guidance activity in RGCs, whereas mutant mice develop PHPV with the presence of a large hyperplastic mass within the vitreous consisting of residual hyaloid vessels, as well as pigmented cells of neural crest origin [96,97]. Interestingly, PHPV caused by *Efna5* deletion is not caused by failed hyaloid regression, but rather due to an alternative mechanism with the excessive recruitment of neural crest and mesodermal cells to the primary vitreous. This makes our finding especially significant, since the development of the primary vitreous takes place in early gestation, starting in weeks 4–6 and thus roughly corresponds to the early timing of week 7 retinal organoids [98]. Additionally, within our dataset, we also observed ATOH7 binding to an enhancer of the related ephrin *EFNA4,* which has been shown to be necessary for the proangiogenic function of VEGF in pathological neovascularization in a mouse model of oxygen-induced retinopathy [99].

Although not identified by pathway databases, we found several other highly regulated axon-related genes overlapping with early RGC-specific markers, including *AFAP1*, *PPP1R1A,* and *ADAM11.* The actin filament-associated protein *AFAP1* was the most significant DEG amongst the reporter cells (Figure 5A,B). *Afap1* is enriched in peripheral ON-parasol RGCs, and a variant within *AFAP1* has been associated with primary open-angle glaucoma in human patients [100]. *PPP1R1A* was associated with an ATOH7-binding site within its promoter region. PPP1R1A is enriched in neural growth cones in mice and may therefore potentially have importance in axonal development [101]. Also, *ADAM11* was associated with an ATOH7-binding site within its promoter. The encoded metalloprotease protein is expressed predominantly in the CNS and is implicated in neural adhesion and axon guidance due to its similarity to *ADAM23* [102]. One novel ATOH7 target candidate is *LDB3*. This gene was found among the top three most significant DEGs amongst ATOH7-positive cells and, interestingly, was also present amongst the RGC markers. This PDZ-LIM protein-encoding gene has to date only been reported in association with myopathies; however, a recent study indicated that alternatively spliced *LDB3* transcripts are broadly expressed in the central and peripheral nervous systems of humans and mice [103].

Therefore, we conclude that the presence of these genes in both the CUT&RUN sequencing data as well as in the reporter-enriched mRNA sequencing dataset suggests a direct involvement of ATOH7 in axon development and guidance. Moreover, most of these target DEGs were also found to be downregulated in both early and late RGC clusters in the scRNA-sequencing data (Appendix A).

### 4.2. ATOH7 Regulates Pluripotency and Differentiation through Notch Signaling

Notch inhibition is a critical step in RPC differentiation towards adult neural cell types as it promotes proliferation and inhibits differentiation [104]. In agreement with previous studies, the Notch signaling pathway was significantly enriched amongst ATOH7 target DEGs, where ATOH7 exerted inhibition on the Notch receptor *NOTCH1* and the bHLH transcription factor *HES6* (Figure 7D) [26,27]. NOTCH1 controls downstream targets from the HES and HEY families, likely causing the observed upregulation of *HES2, HES5*, *HEYL*, and the ATOH7 target gene *HES6* in mutant reporter cells (Appendix A). Interestingly, the Notch-associated bHLH transcription factor *HEY1* was also identified as a target DEG and showed the strongest downregulation amongst Notch signaling genes (Figure 7D, Appendix A), despite being previously reported as a positively regulated NOTCH1 effector in adenoid cystic carcinoma cells [105]. This discrepancy is likely due to the direct activation by ATOH7 in tRPCs and early RGCs.

Previous reports have shown the inhibition of both *Ascl1* and *Neurod4* in the mouse brain by HEY1 [106]. Concordantly, the strong reduction in ATOH7-regulated *HEY1* expression in reporter cell mutants may explain the increased expression of *ASCL1* (Appendix A). The suppression of *ASCL1* through ATOH7-regulated *HEY1* may be a crucial mechanism leading to the terminal division and differentiation of tRPCs, as the misexpression of *Ascl1* in the *Atoh7* retinal cell lineage has been shown to block cell cycle exit without redirecting cell fate [107]. This may also be a contributing reason to the increase in nRPCs in mutant organoids (Figure 6C). Additionally, mutant reporter cells also showed an increased expression of *NEUROD4* (Appendix A). *Neurod4*, together with *Neurod1*, has been implicated in both amacrine cell and rod photoreceptor development [108]. The increased expression of these bHLH transcription factors and increase in cell numbers in RPC, H&As, and PR clusters in mutant organoids is representative of this finding (Figure 6C, Appendix A).

We also identified *KIAA1217* as an nRPC marker gene and a negatively regulated ATOH7 target DEG (Figure 6E). *KIAA1217* has not to our knowledge been described in the retina; however, it has been reported to play a crucial role in hepatocellular carcinoma, promoting cell migration and invasion. Interestingly, *KIAA1217* is associated with the activation and retention of p-STAT3 in the cytoplasm, where it activates the Notch and Wnt/ß-catenin pathways [69]. The downregulation of *KIAA1217* expression could potentially be one of several ways in which ATOH7 inhibits Notch signaling, in addition to regulating *NOTCH1-* and *HES6*-developing progenitors.

### 4.3. Cell Fate Shift in ATOH7-Deficient Human Retinal Organoids

In concordance with previous studies, we observed a decreased number of RGCs and compensatory increase in cell numbers in both H&As and PR clusters in mutant organoids (Figure 6C) [22,23,24,25]. As it has been remarked previously, the interplay between ATOH7 and other co-expressed proneurogenic and neurogenic factors, especially bHLH transcription factors, have a central role in specifying the downstream fate of the multipotent tRPC [27]. Besides the observed ATOH7-mediated inhibition of *HES* family members, the proposed ATOH7-induced HEY1-mediated inhibition of *ASCL1* and *NEUROD4*, as well as the direct activation of well-known RGC-specifying genes, such as *POU4F2*, *ISL1*, and *EYA2*, we saw a significant ATOH7-mediated inhibition of the bHLH factor *NEUROG1* and the forkhead box transcription factor *FOXN4* (Appendix A). The upregulation of *NEUROG1* in mutant reporter cells likely shifts the competence of the developing progenitors towards PRs, as the overexpression of *ngn1* very effectively reprograms chick retinal pigment epithelium into PR precursors [109]. *Foxn4* has been shown to inhibit RGC fate by suppressing *Atoh7* and *Pou4f2* and activating *Ptf1a* expression, thus leading to amacrine cell development [68]. Atoh7 binding and the subsequent downregulation of *Foxn4* were also recently described using CUT&Tag on mouse retina, thus confirming our result [27].

A few ATOH7 target DEGs were identified as PR-specific marker genes in our scRNA-seq dataset, including the well-known PR marker *PDE6B* but also *VXN* and *BCL2L1*. The overexpression of *vxn* in xenopus was shown to promote the differentiation of early-born retinal neurons by interacting and enhancing bHLH transcription factor function [110]. We identified upregulated *VXN* expression in mutant tRPCs and PRs, suggesting the possible promotion of PR differentiation in ATOH7-deficient cells. Conversely, the antiapoptotic effector *BCL2L1* was identified to be preferentially expressed in PR and was upregulated in mutant reporter cells, suggesting that ATOH7 is possibly controlling cell numbers by regulating the survival of PRs. A study has confirmed the relevance of *Bcl2l1* in rod photoreceptors, where loss causes increased susceptibility to light damage [111]. Conversely, *Bcl2l1* was not able to rescue photoreceptor survival in retinal degeneration models [112]. It remains to be investigated whether *BCL2L1* is crucial for early photoreceptor development.

The shift in cell composition towards PR and H&A cell fates may be a direct effect due to the lack of ATOH7, as suggested by its binding to regulatory elements of *NEUROG1*, *FOXN4*, *VXN,* and *BCL2L1*. Alternatively, these findings may be considered a consequence of the reduced number of RGCs in organoids lacking ATOH7.

### 4.4. In Search of ATOH7-Dependent Secreted Proteins Regulating Retinal Development

In addition to cell–cell interactions, it is possible that the survival of *ATOH7*-independent RGCs and the normal development of the retinal vasculature are dependent on paracrine signaling from ATOH7-expressing RGCs. We analyzed a top selection of ATOH7 reporter DEGs (log_2_FC > ±2, *p* < 0.01) annotated to GO:CC “extracellular region”, a term including gene products that are outside the cell and not attached to the cell surface (Figure 8A), such as components of the extracellular matrix (e.g., COL1A1 and VCAN) and signaling molecules (e.g., WNT10A and NOTUM). As already observed by Brodie-Kommit et al., we also found the neuropeptide-encoding *GAL* to be downregulated in ATOH7 reporter cells, showing primarily expression in tRPCs, early RGCs, and H&As. Follow-up experiments in *Gal* mutant mice, however, did not show an effect on hyaloid regression or RGC density [26]. As also proposed by the authors, we saw the ATOH7-dependent expression of *TAC1* in late RGCs, a gene encoding several tachykinin peptide hormones, including substance P, neurokinin A, and neuropeptides K and γ. Of these, substance P has been shown protect RGCs from NMDA-mediated apoptosis, as well as protect from VEGF-induced endothelial barrier breakdown [113].

Furthermore, we observed the ATOH7 binding of two strongly downregulated Wnt-associated genes encoding the Wnt ligand *WNT10A* and Wnt inhibitor *NOTUM.* These genes were expressed in early RGCs and were also present amongst the top significant DEGs found in mutant reporter cells (Figure 5A). *WNT10A* has been implicated in pathological vascular growth in proliferative retinopathy and has also been observed to be upregulated during rod degeneration, activating Wnt signaling and thus protecting PRs from oxidative stress [114,115]. NOTUM, on the other hand, is a Wnt-specific inhibitor that acts by removing an essential palmitoleate moiety from Wnt ligands [116]. Considering the importance of Wnt signaling in both retinal vascularization and neuroprotection, the potential role of these genes in vascular development and RGC survival would merit further investigation [117,118].

We also identified the dysregulation of several growth factors, including the downregulation of late RGC-expressed *FGF5* and *TGFA* and the upregulation of PR-expressed *EGF* in mutant cells. While all three growth factors have observed mitogenic effect on retinal neuroepithelium, FGF5 has been shown to mediate neuroprotection on RGC-5 cells [119,120]. In addition to growth factors, we also detected a strong increase in the expression of the tRPC-expressed BMP regulator *BMPER*, which has been implicated in the negative regulation of pathological retinal revascularization. *Bmper* mRNA and protein expression is downregulated in a mouse model of oxygen-induced retinopathy, and haploinsufficient mice display increased BMP signaling and the resulting revascularization of the hypoxic retina [121].

### 4.5. ATOH7-Associated Disease May Be a Result of Norrin Deficiency

Mutations in the norrin-encoding gene *NDP* cause Norrie disease (OMIM: #310600), an X-linked syndrome characterized by severe ocular malformations, including persistent fetal vasculature, retrolental fibrovascular masses, microphthalmia, and cataracts, as well as a variable occurrence of extraocular clinical manifestations, including sensorineural hearing loss and cognitive impairments [71,122]. Furthermore, isolated *NDP*-associated ocular malformations may also be diagnosed as X-linked FEVR (OMIM: #305390), Coat’s disease (OMIM #300216), retinopathy of prematurity (OMIM: #133780), as well as PHPV [122]. The secreted norrin is an activator of the Wnt/β-catenin signaling pathway and plays a crucial role not only in retinal vascularization but also during the regression of hyaloid vessels [71,123,124,125]. Norrin is expressed in the developing as well as adult neuroretina. *NDP* transcripts have been detected through RNA *in situ* hybridization in outer and inner nuclear layers, as well as the ganglion cell layer in mice [126,127]. On the other hand, alkaline phosphatase reporter experiments in postnatal mice revealed activity spanning vertically throughout the entire neuroretina, suggesting Müller glia as the principal source of NDP [128].

In our study, mutant reporter cells showed a significant decrease in *NDP* expression (Appendix A). We were able to identify a subset of *NDP*-expressing cells mostly localized to nRPCs, followed by predominantly *ATOH7*-positive tRPCs and exclusively ATOH7-positive early RGCs (Figure 8C). Conversely, mutant organoids showed a strong decline in *NDP*-positive RPCs and the absence of expressing RGCs. These results imply an ATOH7-dependant expression of *NDP* in developing progenitors and RGCs, although likely not by ATOH7 directly binding to regulatory elements. Furthermore, *NDP* expression in retinal progenitors has been shown to be regulated by Shh by inducing Gli2 binding to the *Ndp* promoter [129]. Thus, the significant reduction in *NDP*-expressing nRPCs may be secondary to the lack of *SHH*-expressing late RGCs (Appendix A).

An insufficient *NDP* expression could be a contributing factor to the vascular abnormalities, as well as the secondary loss of RGCs in *ATOH7*-deficient retinae. Firstly, as the primary vitreous forms at this early time point in human development, a reduced total NDP availability may have an impact on the formation of the hyaloid vessels [99]. *NDP*-expressing RGCs may have a specific significance, given their proximity to the vitreous and hyaloid. Additionally, in mutant reporter cells, we detected an upregulated expression of *LRP5* and *LGR4*, coding for NDP-associated receptors, which could potentially sequester and further reduce free amounts of NDP in the early retina (Figure 8D, Appendix A). Secondly, NDP has been shown to mediate a neuroprotective role for RGCs by indirectly stimulating the release of neuroprotective factors from Müller glia. This is evidenced by reduced NMDA-mediated apoptosis upon exogenous NDP treatment, as well as by the progressive loss of RGCs in *NDP*-deficient mice [130].

## 5. Conclusions

This study aimed at gaining a better insight into the role of *ATOH7* in early human neuroretinal development. Our data support a role for this bHLH transcription factor in guiding RPC competence towards RGC commitment by regulating Notch signaling, activating RGC-specifying transcription factors, directly inducing axon development and guidance, as well as by inhibiting genes associated with other early-born retinal cell types. We highlight specific candidate target genes that may be directly responsible for causing the underdevelopment or absence of the optic nerve, contributing to the secondary loss of *ATOH7*-independent RGCs, and propagating the vascular anomalies. The results of this study consolidate current knowledge from animal studies and present new insights into human retinal development and possible therapeutic targets for RGC-related and retinal diseases.

## Figures and Tables

**Figure 1 cells-13-01142-f001:**
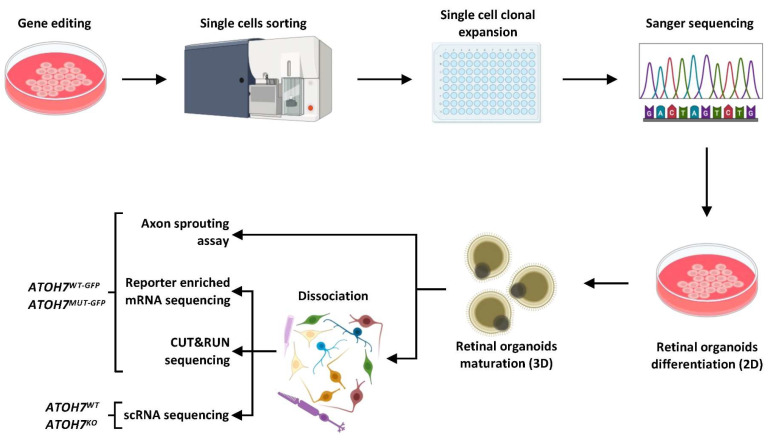
Graphical summary of the study design. Human iPSCs were edited using a CRISPR/Cas9 approach. From the edited cell population, single cells were sorted and expanded in 96-well plates, followed by genotyping by Sanger sequencing. Selected clones were then differentiated to retinal organoids, which were then collected for different experiments. iPSC, induced pluripotent stem cell; WT, wildtype; KO, knockout; CUT&RUN-seq, cleavage under targets and release using nuclease sequencing; scRNA-seq, single-cell RNA sequencing; MUT-GFP, mutant reporter; WT-GFP, wildtype reporter.

**Figure 2 cells-13-01142-f002:**
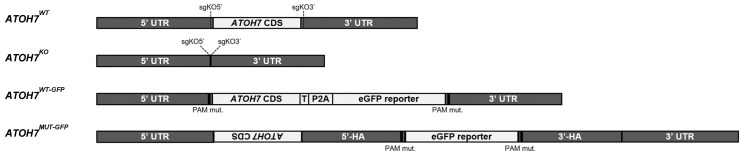
Schematic representation of the iPSC cell lines used in the current study, which include isogenic WT and KO clones (*ATOH7^WT^* and *ATOH7^KO^*), as well as WT and mutant eGFP reporter lines (*ATOH7^WT-GFP^* and *ATOH7^MUT-GFP^*). iPSC, induced pluripotent stem cell; WT, wildtype; KO, knockout; UTR, untranslated region; CDS, coding sequence; T, c-Myc tag; eGFP, enhanced green fluorescent protein; P2A, porcine teschovirus-1 2A self-cleaving peptide; HA, homology arm; sgKO, single-guide RNA for ATOH7 knockout; PAM, protospacer-adjacent motif; MUT-GFP, mutant reporter; WT-GFP, wildtype reporter.

**Figure 3 cells-13-01142-f003:**
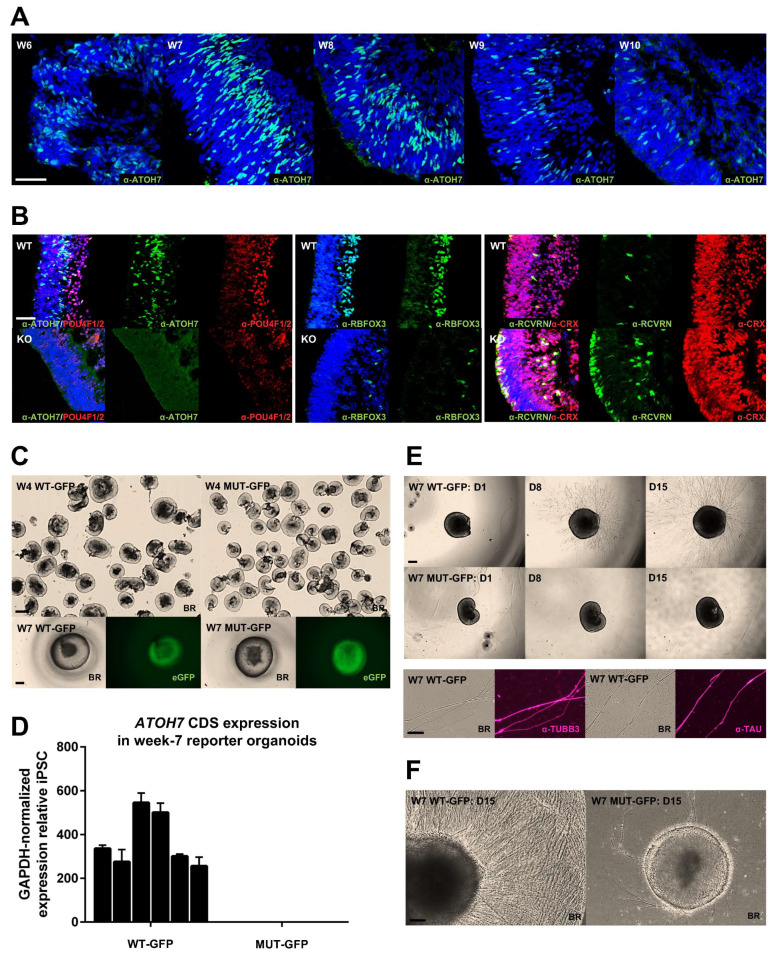
Characterization of retinal organoids derived from gene-edited human iPSCs. (**A**) Confocal laser scanning microscopy imaging of immuno-assayed week 6–10 retinal organoids demonstrating protein expression of ATOH7 and (**B**) comparison of RGC and PR markers in *ATOH7^WT^* (WT) and *ATOH7^KO^* (KO) week 7 retinal organoids. Scale bar represents 50 µm. (**C**) Week 4 retinal organoids derived from *ATOH7^WT-GFP^* and *ATOH7^MUT-GFP^* reporter iPSCs immediately after dissection from 2D differentiation culture. At week 7, clear GFP signals could be detected in reporter iPSC-derived organoids. Scale bars represent 200 µm and 100 µm. (**D**) RT-qPCR targeting *ATOH7* CDS in week 7 retinal organoids derived from reporter iPSCs. The assay was performed on six individual differentiation batches in a triplicate design. The cycle threshold results were normalized to GAPDH levels and are shown relative to expression in iPSCs. (**E**) Week 7 retinal organoids derived from reporter iPSCs were cultured on coverslips coated with Matrigel^®^ for two weeks in order to observe axon sprouting. Immunostaining targeting neuronal tubulin TUBB3 and axon-specific TAU confirmed axon identity. Scale bars represent 200 µm and 100 µm. (**F**) Week 7 retinal organoids were also cultured encapsulated inside Corning^®^ Matrigel^®^ Matrix domes for two weeks, resulting in enhanced axon sprouting and confirming reduced axonogenesis in *ATOH7*-mutant retinal organoids. Scale bar represents 100 µm. BR, brightfield; iPSC, induced pluripotent stem cell; KO, knockout; PR, photoreceptor; RGC, retinal ganglion cell; RT-qPCR, real-time quantitative polymerase chain reaction; MUT-GFP, mutant reporter; WT-GFP, wildtype reporter; WT, wildtype. Scale bars: 50 µm (63X: (**A**,**B**)), 200 µm (4X: (**C**,**E**)), 100 µm (10X: (**C**,**F**)), and 100 µm (20X: (**E**)).

**Figure 4 cells-13-01142-f004:**
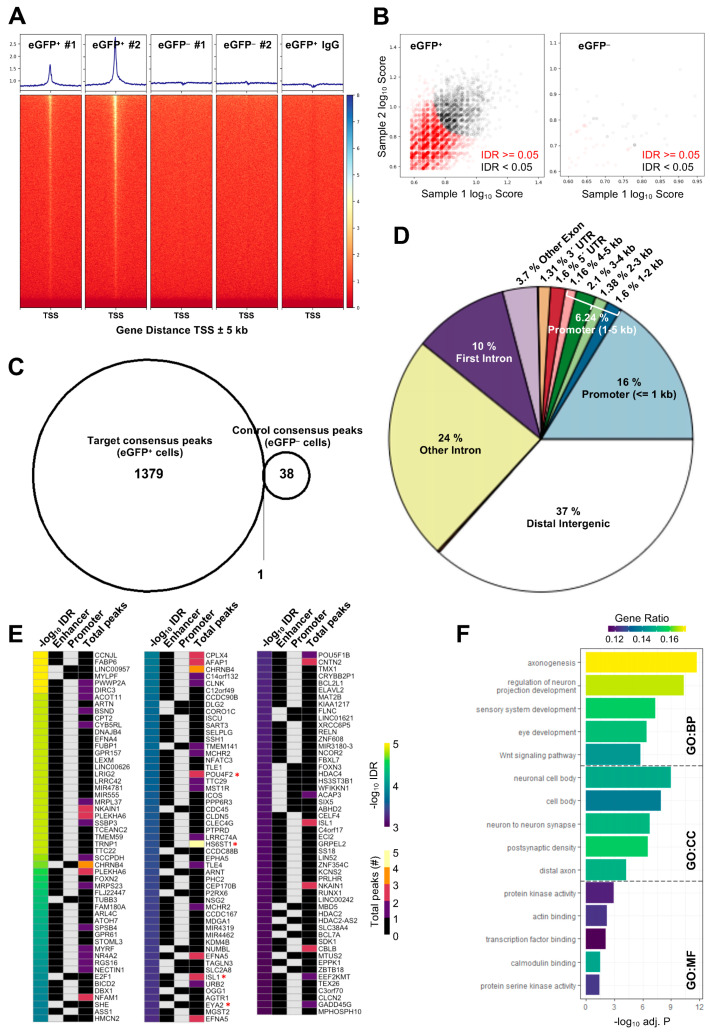
Identification of ATOH7 downstream targets by CUT&RUN-seq, performed on week 7 retinal organoids derived from *ATOH7^WT-GFP^* iPSCs. (**A**) Enrichment of reads within 5 kb of known TSSs acquired from CUT&RUN-seq of anti-ATOH7 antibody-treated eGFP-positive cells (target), eGFP-negative cells (negative control), and non-immune IgG-treated eGFP-positive cells (background control). (**B**) Peaks were determined using the MACS2 algorithm testing enrichment against the background control. Consensus peaks between replicate experiments were identified using the IDR method with a threshold of IDR < 0.05. (**C**) Venn diagram showing number of consensus peaks in target and negative control identified by IDR. The most enriched known motif found amongst target consensus peaks was the bHLH E-box motif modeled by ATOH1 using HOMER. (**D**) Genetic element distribution of target consensus peaks. (**E**) Top target consensus peaks ranked by lowest IDR (heatmap cut-off IDR < 0.001) and their peak-to-gene annotation using ChIP-Enrich. Additionally, the heatmap visualizes whether the annotated peak was assigned as a distal enhancer (>5 kb from TSS) or promoter (±5 kb from TSS), and total target consensus peaks assigned to the gene of interest. * Annotated by multiple peaks and/or discussed more in detail in the Results Section. (**F**) GO enrichment by g:Profiler of annotated genes after reduction using REVIGO, showing the top five most significant terms per GO category according to adjusted *p*-values, based on g:SCS. BP, biological process; CC, cellular compartment; CUT&RUN-seq, cleavage under targets and release using nuclease sequencing; g:SCS, g:Profiler set counts and sizes; GO, gene ontology; IDR, irreproducible discovery rate; MACS2, model-based analysis for ChIP-Seq 2; MF, molecular function; REVIGO, reduce visualize gene ontology; WT-GFP, wildtype reporter; TSS, transcription start site.

**Figure 5 cells-13-01142-f005:**
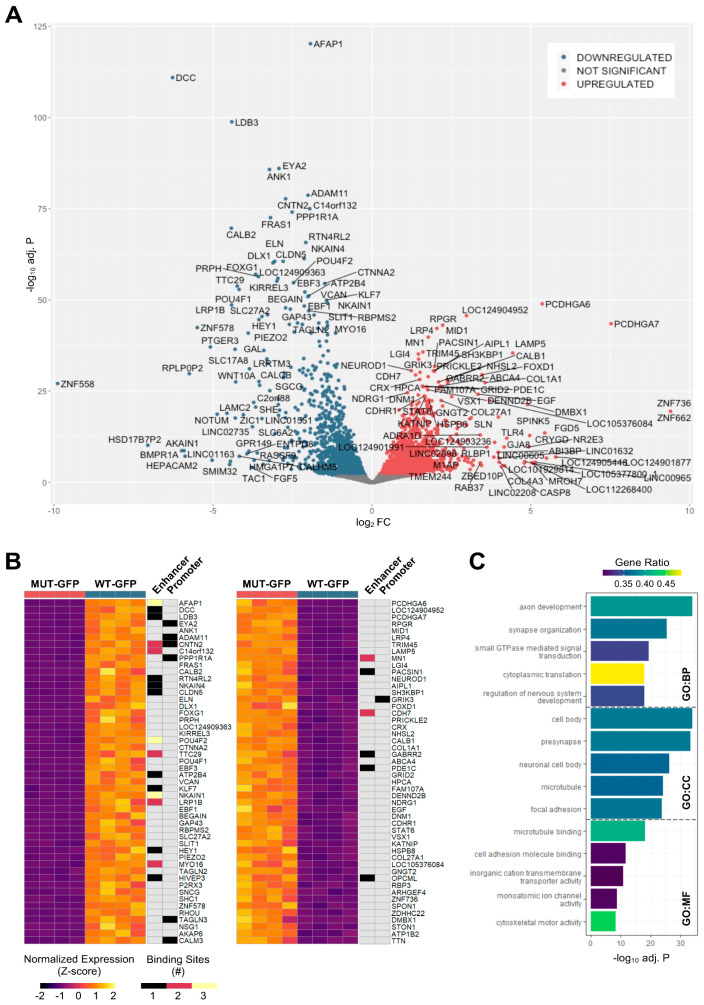
Differentially expressed genes (DEGs) in retinal organoid-derived *ATOH7* reporter cells. (**A**) Reporter cells were isolated from week 7 organoids, sorted by cytometry, and sequenced by mRNA-seq. Differential expression analysis between *ATOH7^MUT-GFP^* and *ATOH7^WT-GFP^* reporter-enriched cells revealed 4037 DEGs, of which the top candidates are illustrated by Volcano using fold change and adjusted *p*-value (Benjamini–Hochberg FDR). (**B**) Heatmaps showing top-ranking DEGs according to adjusted *p*-value, as well as the presence and number of annotated ATOH7 binding sites, as revealed by CUT&RUN-seq. (**C**) GO enrichment by g:Profiler of annotated genes after reduction using REVIGO, showing the top five most significant terms per GO category according to adjusted *p*-values, based on g:SCS. BP, biological process; CC, cellular compartment; CUT&RUN-seq, cleavage under targets and release using nuclease sequencing; DEG, differentially expressed gene; FC, fold change; FDR, false discovery rate; g:SCS, g:Profiler set counts and sizes; GO, gene ontology; REVIGO, reduce visualize gene ontology; MUT-GFP, mutant reporter; WT-GFP, wildtype reporter.

**Figure 6 cells-13-01142-f006:**
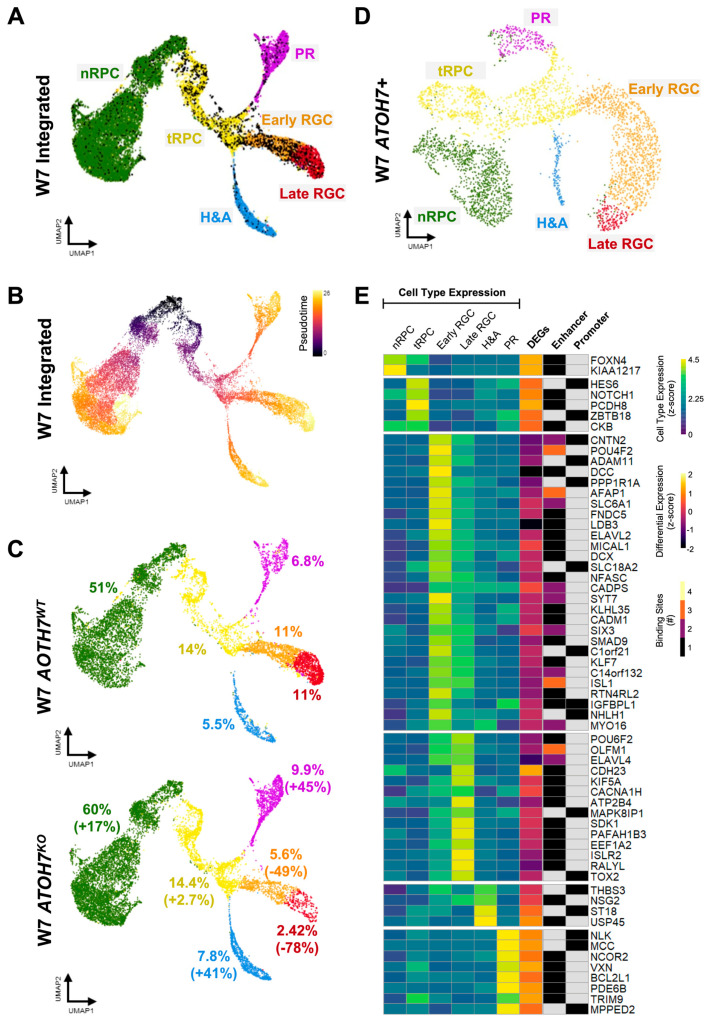
Retinal organoid cell composition and cell type-specific genes regulated by ATOH7. Ten week 7 *ATOH7^WT^* and *ATOH7^KO^* retinal organoids were separately pooled, dissociated, and subsequently analyzed by hybridization-based PFA-fixed scRNA-seq. (**A**) UMAP of defined retinal clusters from integrated *ATOH7^WT^* and *ATOH7^KO^* organoids. The representative clusters are defined as naïve retinal progenitor cells (**nRPCs**), transient retinal progenitor cells (**tRPCs**), early retinal ganglion cells (**early RGCs**), late retinal ganglion cells (**late RGCs**), horizontal and amacrine cells (**H&As**), and photoreceptor cells (**PRs**). (**B**) Pseudotime trajectory analysis starting from *TP53*-positive dividing retinal progenitor cells. (**C**) Separated UMAPs of cells derived from either *ATOH7^WT^* or *ATOH7^KO^* organoids. Percentages represent the proportion of cells in a given cluster compared to the total number of cells. Values in parentheses indicate the relative change in cell proportions compared to WT. (**D**) UMAP of re-clustered *ATOH7*-positive cells derived from *ATOH7^WT^* organoids. (**E**) ATOH7-targeted cell type marker genes, defined as scRNA-seq marker genes (AUC > 0.75) with a significant differential expression (adj. *p*-value < 0.01) between *ATOH7^MUT-GFP^* and *ATOH7^WT-GFP^* cells, based on reporter-enriched mRNA-seq, and at least one binding site discovered by CUT&RUN-seq, where the annotation is defined as either enhancer (>5 kb from TSS) or promoter (±5 kb from TSS). AUC, area under curve; CUT&RUN-seq, cleavage under targets and release using nuclease sequencing; KO, knockout; PFA, paraformaldehyde; scRNA-seq, single-cell RNA sequencing; TSS, transcription start site; UMAP, uniform manifold approximation and projection; DEGs, differentially expressed genes between MUT-GFP and WT-GFP; MUT-GFP, mutant reporter; WT-GFP, wildtype reporter; WT, wildtype.

**Figure 7 cells-13-01142-f007:**
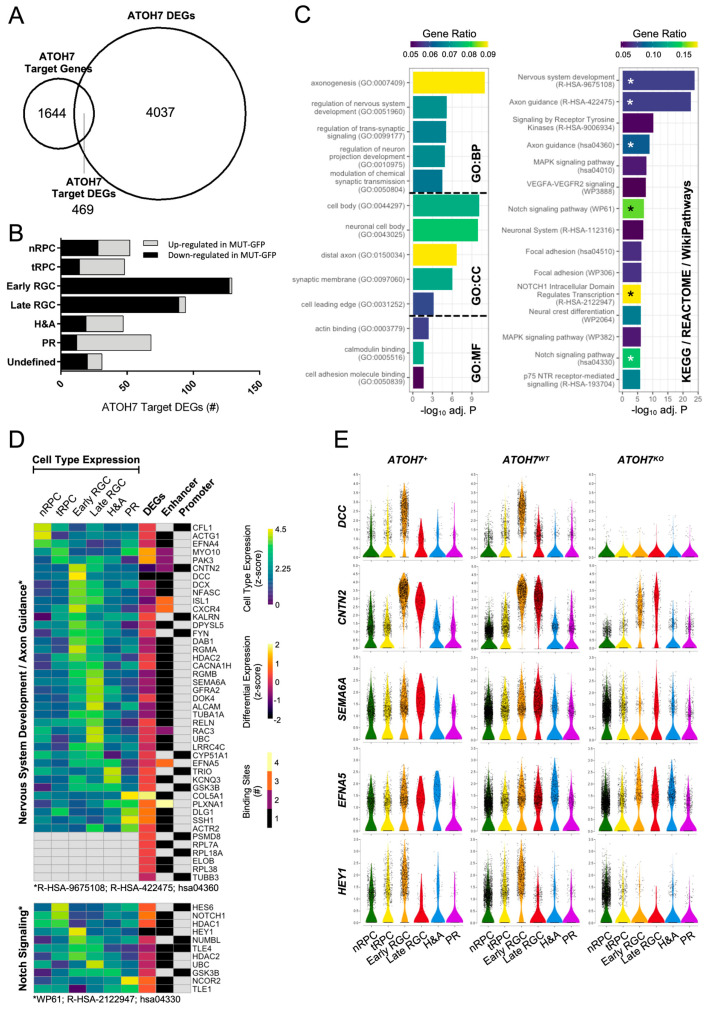
Axon guidance and Notch signaling are enriched amongst ATOH7 target DEGs. Identified ATOH7 target genes and *ATOH7*-associated DEGs in eGFP-enriched reporter cells derived from week 7 retinal organoids were overlapped to identify differentially expressed ATOH7 target genes (ATOH7 target DEGs). (**A**) Venn diagram showing the identification of ATOH7 target DEGs. (**B**) Number of regulated genes per cell type amongst ATOH7 target DEGs, with cell-type specificity defined by scRNA-seq. The representative cell clusters are defined as naïve retinal progenitor cells (**nRPCs**), transient retinal progenitor cells (**tRPCs**), early retinal ganglion cells (**early RGCs**), late retinal ganglion cells (**late RGCs**), horizontal and amacrine cells (**H&As**), and photoreceptor cells (**PRs**). (**C**) Enrichment of GO terms amongst the 469 ATOH7 target DEGs compared to all annotated genes. The analysis was performed in g:Profiler and reduced by REVIGO, showing the top five most significant terms for biological processes (**GO:BP**), cellular compartments (**GO:CC**), and molecular function (**GO:MF**) for ATOH7 target DEGs. Additionally, the enrichment of biological pathway terms from combined KEGG, REACTOME, and WikiPathways databases was performed against all known human genes, showing top 15 terms according to adjusted *p*-value. Asterisks (*) demark significance (adj. *p*-value < 0.05) when including annotated genes only. Multiple correction testing was performed using g:SCS. (**D**) Selection of ATOH7 target DEGs intersecting with top significantly enriched biological pathways. Similar terms from different databases were grouped together. “Nervous System Development” (R-HSA-9675108) and “Axon Guidance” (R-HSA-422475) were grouped together due to an almost complete overlap (98%) of intersecting genes. Heatmaps of annotated genes present relative expression per cell type, based on scRNA-seq data of *ATOH7*-positive cells in *ATOH7^WT^* organoids, differential expression between *ATOH7^MUT-GFP^* and *ATOH7^WT-GFP^* cells, based on reporter-enriched mRNA-seq, and the number of annotated ATOH7-binding loci identified by CUT&RUN-seq, where the annotation is defined as either enhancer (>5 kb from TSS) or promoter (±5 kb from TSS). (**E**) Expression distribution of selected genes in cell type-specific scRNA-seq clusters. Violin plots show raw expression distribution for cells originating from *ATOH7^WT^* organoids, *ATOH7^KO^* organoids, and from the *ATOH7*-expressing (*ATOH7^+^)* cells, re-clustered from the *ATOH7^WT^* organoids. Each expressing cell is marked by a black point. CUT&RUN-seq, cleavage under targets and release using nuclease sequencing; EVR, exudative vitreoretinopathy; g:SCS, g:Profiler set counts and sizes; GO, gene ontology; KO, knockout; REVIGO, reduce visualize gene ontology; PHPV, persistent hyperplastic primary vitreous; scRNA-seq, single-cell RNA sequencing; DEGs, differentially expressed genes between MUT-GFP and WT-GFP; MUT-GFP, mutant reporter; WT-GFP, wildtype reporter; TSS, transcription start site; WT, wildtype.

**Figure 8 cells-13-01142-f008:**
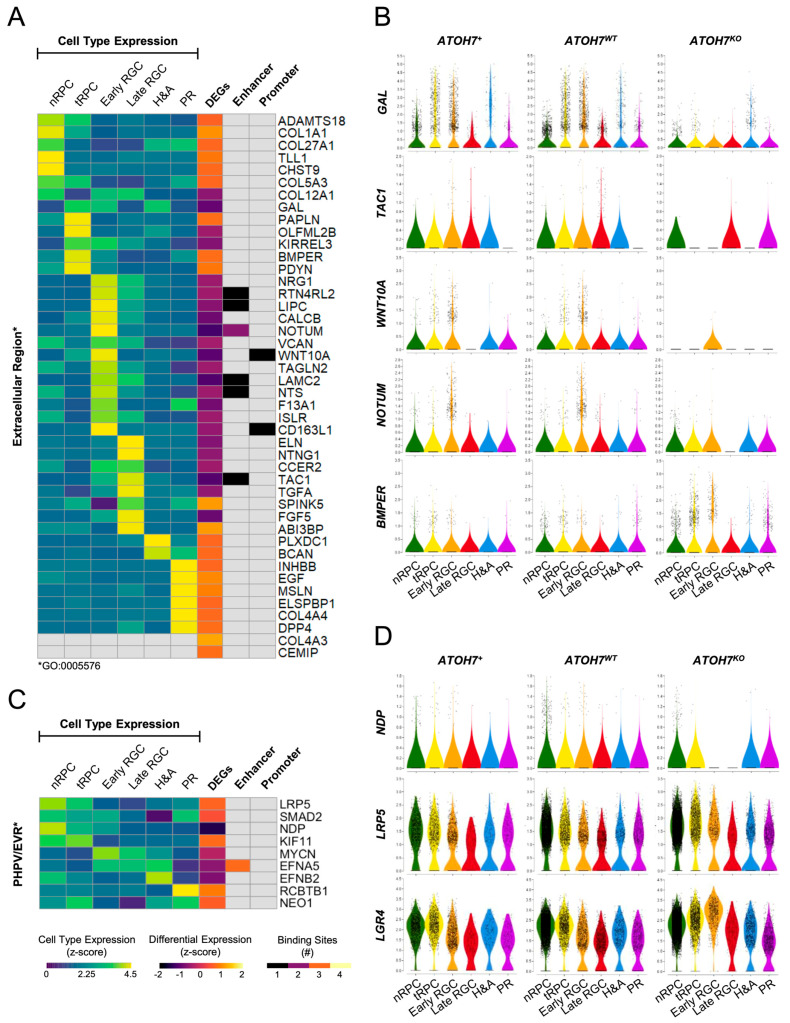
ATOH7 regulates genes encoding Wnt effectors and other secreted proteins. (**A**) Selection of *ATOH7* reporter DEGs (log_2_FC > ±2, FDR < 0.01) encoding extracellular/secreted proteins. Heatmap of annotated genes present relative expression per cell type, based on scRNA-seq data, differential expression between *ATOH7^MUT-GFP^* and *ATOH7^WT-GFP^* cells, based on reporter-enriched mRNA-seq, and the number of annotated ATOH7-binding loci identified by CUT&RUN-seq, where the annotation is defined as either enhancer (>5 kb from TSS) or promoter (±5 kb from TSS). The representative cell clusters are defined as naïve retinal progenitor cells (**nRPCs**), transient retinal progenitor cells (**tRPCs**), early retinal ganglion cells (**early RGCs**), late retinal ganglion cells (**late RGCs**), horizontal and amacrine cells (**H&As**), and photoreceptor cells (**PRs**). (**B**) Expression distribution of selected genes encoding secreted proteins in cell type-specific scRNA-seq clusters. Violin plots show raw expression distribution in cells originating from *ATOH7^WT^* organoids, *ATOH7^KO^* organoids, and from the *ATOH7* expressing (***ATOH7^+^****)* cells, re-clustered from the *ATOH7^WT^* organoids. Each expressing cell is marked by a black dot. (**C**) Heatmap including the selection of ATOH7 reporter DEGs (log_2_FC > ±2, FDR < 0.01) associated with PHPV/EVR in patients (*LRP5*, *NDP*, *KIF11*, and *RCBTB1*) and/or in vivo (*LRP5*, *NDP*, *KIF11*, *EFNA5*, *EFNB2*, and *NEO1*) or in silico (*SMAD2* and, *MYCN*) according to a literature search [70,71,72,73,74,75,76,77,78,79,80,81,82,83,84,85,86,87,88,89,90,91,92,93,94,95,96]. (**D**) Expression distribution of genes encoding *NDP* and *NDP*-associated receptors in cell type-specific scRNA-seq clusters. **CUT&RUN-seq**, cleavage under targets and release using nuclease sequencing; **EVR**, exudative vitreoretinopathy; **KO**, knockout; **PHPV**, persistent hyperplastic primary vitreous; **scRNA-seq**, single-cell RNA sequencing; **DEGs**, differentially expressed genes between MUT-GFP and WT-GFP; **MUT-GFP**, mutant reporter; **WT-GFP**, wildtype reporter; **TSS**, transcription start site; **WT**, wildtype.

**Table 1 cells-13-01142-t001:** HOMER known motif enrichment in sequences of target consensus peaks. Top 5 known motifs enriched amongst sequences of CUT&RUN target consensus peaks ranked according to Benjamini–Hochberg FDR. BG, background; FDR, false discovery rate; HOMER, hypergeometric optimization of motif enrichment.

Rank	Known Motif	Name	*p*-Value	FDR	% Targets	% BG
1	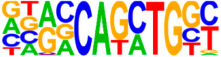	Atoh1(bHLH)/Cerebellum-Atoh1-ChIP-Seq(GSE22111)/Homer	1 × 10^−402^	0.000	66.6	15.3
2	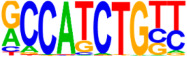	NeuroD1(bHLH)/Islet-NeuroD1-ChIP-Seq(GSE30298)/Homer	1 × 10^−396^	0.000	57.1	10.4
3	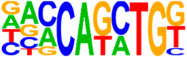	BHLHA15(bHLH)/NIH3T3-BHLHB8.HA-ChIP-Seq(GSE119782)/Homer	1 × 10^−368^	0.000	69.9	19.0
4	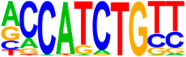	NeuroG2(bHLH)/Fibroblast-NeuroG2-ChIP-Seq(GSE75910)/Homer	1 × 10^−340^	0.000	68.1	19.2
5	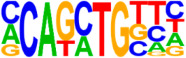	Twist2(bHLH)/Myoblast-Twist2.Ty1-ChIP-Seq(GSE127998)/Homer	1 × 10^−333^	0.000	73.3	23.5

**Table 2 cells-13-01142-t002:** Top 5 HOMER de novo motifs enriched amongst sequences of CUT&RUN target consensus peaks ranked according to *p*-value. Best matching known motifs are presented for each de novo sequence along with a motif match score (0–1), as well as alternative motif matches with matching scores in parentheses. BG, background; HOMER, hypergeometric optimization of motif enrichment.

Rank	De Novo Motif	*p*-Value	% Targets	% BG	Best Motif Match	Score	Similar Motif Matches
**1**	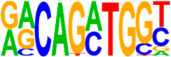	1 × 10^−407^	61.7	12.3	NeuroD1(bHLH)/Islet-NeuroD1-ChIP-Seq(GSE30298)/Homer	0.976	NeuroG2 (0.97); Atoh1 (0.95); NEUROD1 (0.95); NEUROG2 (0.94); Olig2 (0.93); Twist2 (0.91); BHLHA15 (0.91); Ascl1 (0.90)
**2**	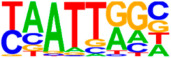	1 × 10^−85^	47.0	22.9	En1(Homeobox)/SUM149-EN1-ChIP-Seq(GSE120957)/Homer	0.942	DLX2 (0.94); Lhx8 (0.93); VSX1 (0.93); Lhx2 (0.92); DLX1 (0.92); DLX5 (0.92); Lhx1 (0.92); DRGX (0.92); LHX9 (0.92)
**3**	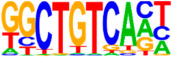	1 × 10^−66^	35.2	16.0	MEIS2/MA0774.1/Jaspar	0.922	Achi (0.90); MEIS3 (0.89); hth (0.88); vis (0.88); Pknox2 (0.87); Vis (0.87); Tgif1 (0.87); Hth (0.87)
**4**	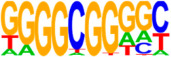	1 × 10^−49^	11.5	2.80	Sp2(Zf)/HEK293-Sp2.eGFP-ChIP-Seq(Encode)/Homer	0.963	KLF15 (0.94); Sp1 (0.94); GC-box (0.93); KLF1 (0.90); Sp5 (0.90); KLF5 (0.89); btd (0.88); Klf7 (0.88); Bcl6b (0.88)
**5**	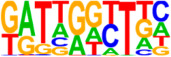	1 × 10^−21^	8.12	2.82	DUX4(Homeobox)/Myoblasts-DUX4.V5-ChIP-Seq(GSE75791)/Homer	0.673	Duxbl (0.67); NFYA (0.66); DUXA (0.65); ONECUT1 (0.65); NFY (0.65); NFYC (0.65); CCAAT-box (0.65); ceh-48 (0.64)

## Data Availability

The original data presented in this study are openly available from Zenodo at 10.5281/zenodo.11473148.

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
