# Peer review of "Identification and Characterization of ATOH7-Regulated Target Genes and Pathways in Human Neuroretinal Development"

_cells, 2024, doi:10.3390/cells13131142_

Round 1

Reviewer 1 Report

Comments and Suggestions for Authors

Atac and colleagues provide a very detailed and comprehensive analysis of gene expression changes in human iPS cell-derived neuroretinal organoids that are linked to the regulatory function of the ATOH7 (MATH5) transcription factor. ATOH7 is critical for retinal ganglion cell specification and development since loss of a functional gene leads to a retina with under-developed RGCs and abnormal optic nerve. They do this by using essentially 3 approaches, including evaluation of retinal organoids with WT or KO ATOH7, CUT&RUN identification of ATOH7 binding sites, and scRNA-Seq of organoid cells comparing WT and KO or ATOH7 mutant organoids.

The findings are well-described, and the interpretation of the data is sound and reasonable. Loss of ATOH7 results in minimal RGC development and reduced neurite formation of the organoids. Of great interest is the apparent switch of 7-week organoids to different cell fates, suggesting a negative regulatory role for ATOH7 in suppressing, perhaps, default specification pathways. ATOH7 itself is involved heavily in the regulation of genes involved in axon development and guidance, as well at Notch signaling. These pathways appear to be corroborated by the combined CUT&RUN and scRNA-Seq approaches.

Surprisingly, or perhaps not so much, there is a lack of concordance of DEG identified in this study and two other similar studies using similar technologies. It is relevant to note that the two other studies also showed a lack of concordance with each other. Similar high variability has also been described in scRNA-seq and microarray studies even when the same experimental model is used, suggesting that these methods are still prone to technical variation or that minute experimental differences can result in highly variable results. Overall, this work is really exciting and adds important knowledge to the developmental function of ATOH7.

Reviewer 2 Report

Comments and Suggestions for Authors

The manuscript by Atac et al., presents novel findings that demonstrate transcriptional targets of ATOH7 in retinal organoids. While Atoh7 targets have been identified other model systems, this is the first report using retinal organoids, and adds significantly to the body of literature concerning the role of ATOH7 in retinal ganglion cell (RGC) differentiation, maintenance, and secondary effects on other retinal cell types. The studies are well designed with appropriate controls, and the methods are clear and include information of the number for organoids or cells used in each experiment. I have detailed three main points that should be addressed before publication. 

1)        The authors claim, based on their results from bulk RNAseq, that ATOH7 is directly involved axon development and guidance (lines 745-755).  The authors data clearly shows that there are fewer RGCs in ATOH7 mutant organoids and given this result, one might expect that there would be reduced expression of genes required for the sprouting and guidance of RGC axons. Ie. If are fewer cells and fewer axons, there would be fewer of the axon guidance molecules expressed. Stating That ATOH7 is directly involved is thus an overstatement. The identification of an axon development or guidance genes from the target gene set would be more informative about whether ATOH7 is directly involved in axon development. This should be addressed in the discussion.

2)        On a similar note, the up-regulation of genes found in photoreceptors and other later born retinal neurons is not really surprising or particularly novel. It has been well established that ATOH7 is required for the timing of cell cycle exit of retinal progenitor cells, with earlier exiting cells more likely to differentiate into RGCs and later exiting cells more likely to become interneural or photoreceptor cell types. Thus the increase in photoreceptor genes likely represents a secondary effect resulting from the altered composition of cell types in the organoid, as opposed to a direct effect on specific genes.  This is apparent as few dysregulated photoreceptor genes are amongst the ATOH7 gene target set (lines 539-540). The discussion should have a paragraph detailing potential directs vs indirect effects of ATOH7 mutation. 

3)        While mutations in ATOH7 are associated with PHPV/EVR, only one direct target gene related to these disease states was uncovered in their data sets. I presume retinal organoids do not have vasculature? Is ATOH7 expressed in retinal vasculature, or is the persistence of fetal vasculature due to ATOH7 mutation secondary to its loss in retinal neurons?  The (presumed) lack of vasculature in retinal organoids could help explain the lack of targets identified.  It has also been noted that there a relatively low level of overlap with target genes identified in previous studies that use animal models. Could the presence of blood vessels in eyes studied in the animal models contribute to this lack of overlap between data sets? 

Author Response

Dear Reviewer,
Please see the attachement for our response to the comments.
